# Asymmetry of movements in CFTR's two ATP sites during pore opening serves their distinct functions

**Ben Sorum[1], Beáta Töröcsik[1,2], László Csanády[1,2]***

[1]Department of Medical Biochemistry, Semmelweis University, Budapest, Hungary; [2]MTA-SE Ion Channel Research Group, Semmelweis University, Budapest, Hungary

**Abstract** CFTR, the chloride channel mutated in cystic fibrosis (CF) patients, is opened by ATP binding to two cytosolic nucleotide binding domains (NBDs), but pore-domain mutations may also impair gating. ATP-bound NBDs dimerize occluding two nucleotides at interfacial binding sites; one site hydrolyzes ATP, the other is inactive. The pore opens upon tightening, and closes upon disengagement, of the catalytic site following ATP hydrolysis. Extent, timing, and role of non-catalytic-site movements are unknown. Here we exploit equilibrium gating of a hydrolysis-deficient mutant and apply Φ value analysis to compare timing of opening-associated movements at multiple locations, from the cytoplasmic ATP sites to the extracellular surface. Marked asynchrony of motion in the two ATP sites reveals their distinct roles in channel gating. The results clarify the molecular mechanisms of functional cross-talk between canonical and degenerate ATP sites in asymmetric ABC proteins, and of the gating defects caused by two common CF mutations.

DOI: https://doi.org/10.7554/eLife.29013.001

*For correspondence:
csanady.laszlo@med.semmelweis-univ.hu

## Introduction

Cystic Fibrosis (CF) is an incurable, devastating inherited disease caused by mutations in the CF Transmembrane Conductance regulator (CFTR) chloride ion channel (*Riordan et al., 1989*). A large number of mutations identified in CF patients scatter throughout the entire protein sequence and cause defects in protein synthesis, maturation, stability, channel gating, and/or anion permeation through the open channel pore (*O'Sullivan and Freedman, 2009*). In recent clinical trials small molecule drugs that target the CFTR protein and rectify either processing ('correctors') or gating defects ('potentiators') have proven beneficial for some CF patients carrying specific mutations (*Ramsey et al., 2011*), underscoring the importance of understanding the structure and mechanism of this disease-associated protein at a molecular level.

CFTR belongs to the large family of ATP-Binding Cassette (ABC) proteins (*Riordan et al., 1989*), most of which function as active transporters (*Dean and Annilo, 2005*). A conserved molecular mechanism that involves ATP binding and hydrolysis drives opening and closing (gating) of the CFTR pore and substrate translocation through other ABC proteins (*Li et al., 1996*). The exceptional resolution of single-channel patch-clamp recordings makes CFTR a model ABC protein eminently suitable for molecular-level biophysical studies.

ABC proteins are built from two homologous halves each containing a transmembrane domain (TMD) and a cytosolic nucleotide binding domain (NBD). In CFTR these two canonical halves are linked by the unique regulatory (R) domain which must be either phosphorylated by cyclic AMP-dependent protein kinase (PKA) (*Anderson et al., 1991*), or deleted (*Figure 1A*) (*Csanády et al., 2000*) to allow channel gating. The highly conserved NBD fold comprises an alpha/beta 'head' subdomain harboring the Walker A and B motifs, and an alpha-helical 'tail' subdomain which contains the ABC-specific 'signature' sequence (*Hung et al., 1998*). Binding of ATP to the Walker motifs in

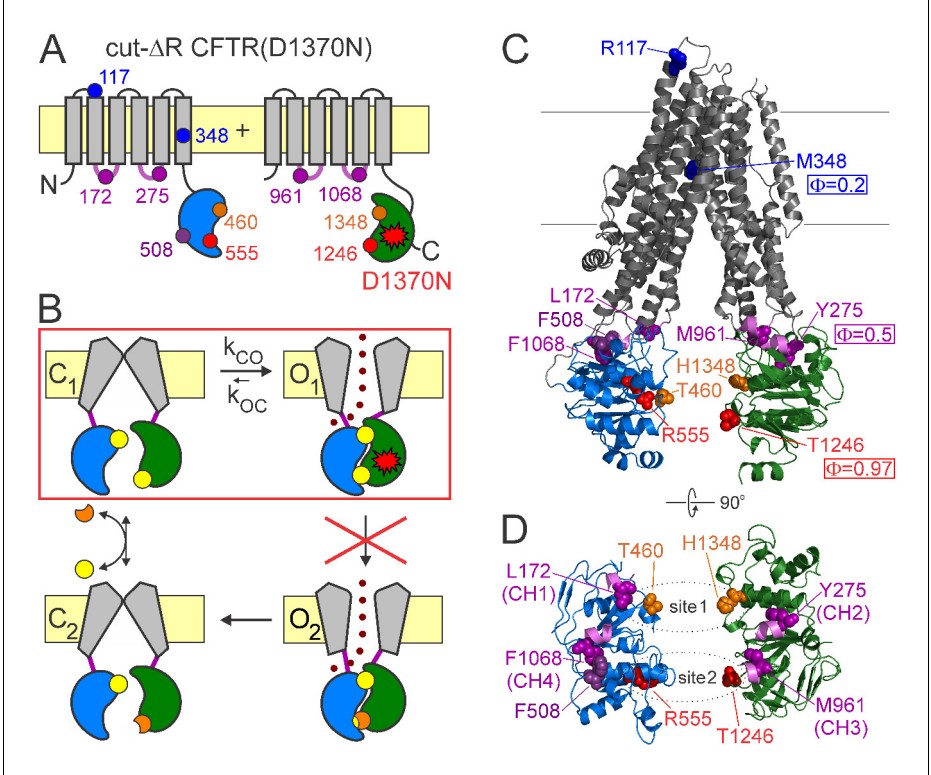

**Figure 1.** CFTR domain topology, gating mechanism, and localization of target positions. (**A**) Domain topology of the CFTR cut-ΔR(D1370N) background construct. TMDs, *gray*; NBD1, *blue*; NBD2, *green*; intracellular loops containing coupling helices (CH1-4, from left to right), *violet*; target positions, *colored circles*. *Red star* denotes mutation D1370N. (**B**) Cartoon gating cycle of WT CFTR; and two-state equilibrium gating (*red box*) in saturating ATP of the background construct in which the D1370N mutation (*red star*) disrupts ATP hydrolysis (*red cross*). ATP, *yellow circles*; ADP, *orange crescents*, *upper* ATP binding site represents non-catalytic site 1, *lower* site represents catalytic site 2. (**C–D**) Target positions highlighted in colored spacefill on the cryoEM structure of dephosphorylated closed human CFTR (PDBID: 5UAK). (**C**) Full-length structure with all target positions shown, color coding as in A. Φ values for positions 1246, 348, and 275 are taken from (*Sorum et al., 2015*). (**D**) Only NBDs and coupling helices (CH1-4) shown, viewed from an angle orthogonal to the membrane. *Dotted ellipses* identify ATP sites 1 and 2.

DOI: https://doi.org/10.7554/eLife.29013.002

each of the two NBDs promotes their association into a tight head-to-tail dimer that occludes the two bound ATP molecules at interfacial binding sites formed by the Walker motifs of one NBD and the signature motif of the other (*Smith et al., 2002*). Such NBD dimers are extremely stable but dissociate following ATP hydrolysis (*Moody et al., 2002*; *Smith et al., 2002*). Formation/disruption of the NBD dimer, communicated to the TMDs through an interface formed by four short coupling helices (CH1-4, *Figure 1A*, *violet loops*), drives the major conformational rearrangements of the substrate translocation pathway (*Locher, 2009*). Upon formation of the NBD dimer, the CFTR anion pore opens to a 'burst' state interrupted by brief (~10 ms) 'flickery' closures, and upon dimer disruption following ATP hydrolysis at the consensus site ('catalytic site 2', formed by NBD2 Walker motifs and NBD1 signature sequence) the channel returns to a long-lived (~1 s) 'interburst' closed state (*Figure 1B*; [*Vergani et al., 2005*]). The closed-pore CFTR structure is inward facing (*Zhang and Chen, 2016*; *Liu et al., 2017*), whereas the open-pore structure, with dimerized NBDs, likely resembles the outward-facing TMD conformation of ABC proteins. In a subset of ABC transporters, including CFTR, only one of the two composite sites is catalytically active, whereas the other contains non-canonical substitutions at several key residues that impair ATPase activity. In CFTR that degenerate ATP site ('non-catalytic site 1', formed by NBD1 Walker motifs and NBD2 signature sequence), keeps ATP bound but unhydrolyzed throughout several gating cycles (*Aleksandrov et al., 2002*;

*Basso et al., 2003*). The precise range of gating-related movements in non-catalytic site 1 is unclear (*Tsai et al., 2010b*; *Szollosi et al., 2011*; *Chaves and Gadsby, 2015*), but mutations here impair ATP hydrolysis at the catalytic site 2 (*Ramjeesingh et al., 1999*), affect channel gating (*Tsai et al., 2010a*; *Csanády et al., 2013*), and may cause CF (e.g., G1349D).

Resolving the relative timing of motions in various protein regions allows the molecular forces that drive protein conformational rearrangements to be understood. In particular, the structural organization of the transition (T) state for channel opening may shed light on the sources of molecular strain that underly the large Gibbs energy of this highest-energy intermediate conformation, which rate limits the overall transition (*Csanády et al., 2006*). Such T-state structures cannot be studied by standard structural approaches, as their life times fall into the sub-microsecond range (*Chakrapani and Auerbach, 2005*). However, relative timing of motions of a given channel position is reported by its $\Phi$ value, the slope of a log-log plot of the opening rate constant ($k_{CO}$) as a function of the closed-open equilibrium constant ($K_{eq} = k_{CO}/k_{OC}$) for a series of point substitutions (Brønsted plot). For the opening step of a simple idealized two-state channel with a single intervening T state, $\Phi$ is a measure of how far the conformation of the position has progressed along the reaction coordinate in the T state ($0 \leq \Phi \leq 1$). For a position that moves very early, and has therefore reached its open-like conformation in the T state ($\Phi = 1$), perturbations affect T-state and open-state stabilities to similar extents, and thus impact only opening but not closing rate. At the other extreme, for a position that moves very late, and is still in its closed-like conformation in the T-state ($\Phi = 0$), perturbations affect closing but not opening rate (*Auerbach, 2007*). In reality, the overall closed-to-open conformational change of large ion-channel proteins is best described by a sequence of conformational steps across a chain of high-energy intermediary states (the 'transition-state ensemble'), and the $\Phi$ value of a position reports whether its perturbations affect early (large $\Phi$) or late (small $\Phi$) steps within that chain of events. However, on the plausible assumption that perturbation of a position affects the microscopic step during which the position undergoes a rearrangement, but not steps during which it remains static, the $\Phi$ value is reasonably interpreted to reflect relative timing of movements even in such cases (*Zhou et al., 2005*). The terms 'early' and 'late' will be used here to reflect sequentiality of movements based on this interpretation.

Because $\Phi$ value analysis requires equilibrium (*Csanády, 2009*), it can be applied to CFTR pore opening only in the presence of a background mutation (*Figure 1A–B*, *red star*) that disrupts ATP hydrolysis at catalytic site 2 (*Figure 1B*, *red cross*), thus reducing gating in saturating ATP to a simple closed-open equilibrium ($C_1 \longleftrightarrow O_1$; *Figure 1B*, *red box*). These conditions are satisfied in a background construct (*Sorum et al., 2015*) in which mutation of the NBD2 Walker B aspartate disrupts ATP hydrolysis and removal of the R domain obviates the need for prior phosphorylation (cut-$\Delta$R (D1370N); *Figure 1A*). In this background, $\Phi$ value analysis detected a clear temporal gradient in opening-related movements which spreads along the longitudinal, cytoplasmic to extracellular, protein axis from catalytic site 2 towards the pore (*Figure 1C*): in the T state catalytic site 2 is already dimerized ($\Phi \sim 1$ for position 1246), the NBD-TMD interface is on the move ($\Phi \sim 0.5$ for position 275), whereas the pore is still shut ($\Phi \sim 0.2$ for position 348), suggesting strain in the region of the main CF locus (*Sorum et al., 2015*).

Here we exploit the same background construct to compare the roles of the two ATP binding sites in pore opening, and find markedly asynchronous movements. An outline of the transition-state structure based on $\Phi$ values of eleven positions at four different levels along the longitudinal protein axis, from the two ATP binding sites to the extracellular surface, suggests distinctly different energetic roles for the two sites in supporting channel gating. This provides the first mechanistic clues for understanding the division of tasks and functional cross-talk between the catalytic and the non-catalytic ATP binding site in asymmetric ABC proteins. The findings also clarify the molecular mechanisms of the CFTR channel gating defects caused by two common CF causing mutations.

## Results

### Longitudinal $\Phi$-gradient extends from the catalytic ATP binding site to the extracellular surface

Early movement within catalytic site 2 was suggested by the large $\Phi$ value for position 1246 in the Walker A motif of NBD2 (*Sorum et al., 2015*). To test timing of movements at the opposing NBD1

surface of catalytic site 2, we targeted position 555 (*Figure 1A,C–D*, *red*) near the NBD1 signature sequence: the R555 and T1246 side chains span the site 2 interface (*Figure 1D*, *lower dotted ellipse*) to form a hydrogen bond in open CFTR channels (*Vergani et al., 2005*). Three tested arbitrary substitutions (see Materials and methods) of the native arginine at position 555 of our background construct (glutamine, alanine, cysteine) all dramatically lowered channel activity (*Figure 2A*): open probability ($P_o$) in saturating (10 mM) ATP was reduced by up to 10-fold (*Figure 2D*). This graded reduction in the closed-open equilibrium constant ($K_{eq}$) was caused predominantly by mutational effects on mean interburst durations ($\tau_{ib}$) which were prolonged by up to 10-fold (*Figure 2C*), reporting large reductions in channel opening rate ($k_{CO} = 1/\tau_{ib}$). Importantly, since 10 mM ATP remained saturating for all of the mutants (*Figure 2—figure supplement 1B*, *red bars*), this reduced rate $k_{CO}$ indeed reflects slowing of the $C_1 \rightarrow O_1$ step (*Figure 1B*). In contrast, mean open burst durations ($\tau_b$) were shortened by less than twofold (*Figure 2B*), reporting only modest effects of the mutations on channel closing rate ($k_{OC} = 1/\tau_b$). Correspondingly, the slope of the Brønsted plot (*Figure 2E*) revealed a high value of $\Phi = 0.84 \pm 0.04$ for position 555, similar to that of position 1246 ($\Phi = 0.97 \pm 0.19$; [*Sorum et al., 2015*]), indicating very early movement of both sides of catalytic site 2 during pore opening.

To further expand the longitudinal axial distance spanned by our target positions, we also studied position 117 (*Figure 1A,C*, *blue*) in the first extracellular loop, where mutations affect gating and are associated with CF (*Sheppard et al., 1993*; *Yu et al., 2016*). Beside a modest reduction in unitary conductance due to the removal of the positive charge of the native arginine, substitution of glutamine, alanine, histidine, or cysteine into position 117 all robustly reduced $P_o$ (*Figure 2F*), by up to ~30 fold (*Figure 2I*). However, compared to the position 555 mutants, the kinetic pattern of gating of the position 117 mutants was strikingly different (cf., *Figure 2F and A*): the reduced $P_o$ in this case reflected a parallel reduction in mean open burst durations, by >50 fold for the histidine and cysteine substitutions (*Figure 2G*), whereas mean interburst durations remained largely unaffected (*Figure 2H*). Thus, perturbations at position 117 selectively affect channel closing, but not opening, rate, yielding a Brønsted plot with a slope of essentially zero (*Figure 2J*). The extremely low value of $\Phi = 0.05 \pm 0.02$ for position 117 implies very late movement of this extracellular position during pore opening, indicating that the conformational wave that begins in catalytic site 2 ends at the extracellular surface.

Because accurate estimation of opening rate requires correct estimation of the number of active channels in the patch, which is difficult when the $P_o$ is small, even with very long recordings, for low-$P_o$ mutants like those in *Figure 2A and F* channel numbers were estimated at the end of each experiment by exposure to 2'-deoxy-ATP (dATP) or 2'-deoxy-$N^6$-(2-phenylethyl)-ATP (P-dATP) which markedly stimulated channel open probability (*Figure 2—figure supplement 2*). Of note, for all mutants studied here the reconstructed interburst $\longleftrightarrow$ burst sequences obtained in saturating ATP (cf., *Figure 2—figure supplement 1*) were reasonably described by a two-step process, as confirmed by dwell-time analysis (see example histograms for 117 and 555 position mutants in *Figure 2—figure supplement 3* and *Figure 2—figure supplement 4*).

## Movements in the non-catalytic ATP binding site are markedly delayed during pore opening

To address timing of movements in non-catalytic site 1, we first targeted positon 1348 in the NBD2 signature sequence (*Figure 1A,C–D*, *orange*). Replacement of the native histidine at position 1348 with an alanine increased $P_o$, whereas substitutions by methionine, tyrosine, or glutamate decreased it (*Figure 3A,D*). In both cases however, changes in $P_o$ reflected simultaneous changes in both mean burst and interburst durations (*Figure 3B,C*; cf., histograms in *Figure 3—figure supplement 1*): that is, shortened bursts typically coupled with prolonged interbursts (H1348E/Y; *Figure 3A–C*), or prolonged bursts coupled with shortened interbursts (H1348A; *Figure 3A–C*). Thus, perturbations at position 1348 alter both opening and closing rates, yielding a Brønsted plot with an intermediate slope (*Figure 3E*): the value of $\Phi = 0.42 \pm 0.04$ stands in stark contrast to $\Phi \sim 1$ for catalytic site 2, and indicates delayed motion at position 1348, relative to site 2.

To target the opposing, NBD1 surface of non-catalytic site 1, we chose position 460 in the NBD1 Walker A motif, corresponding to position 1246 in ATP site 2 (*Figure 1A,C–D*, *orange*). Positions 460 and 1348 flank non-catalytic site 1 (*Figure 1D*, *upper dotted ellipse*) from two sides. Relative to the native threonine at positions 460, the alanine, serine, and proline substitutions markedly reduced

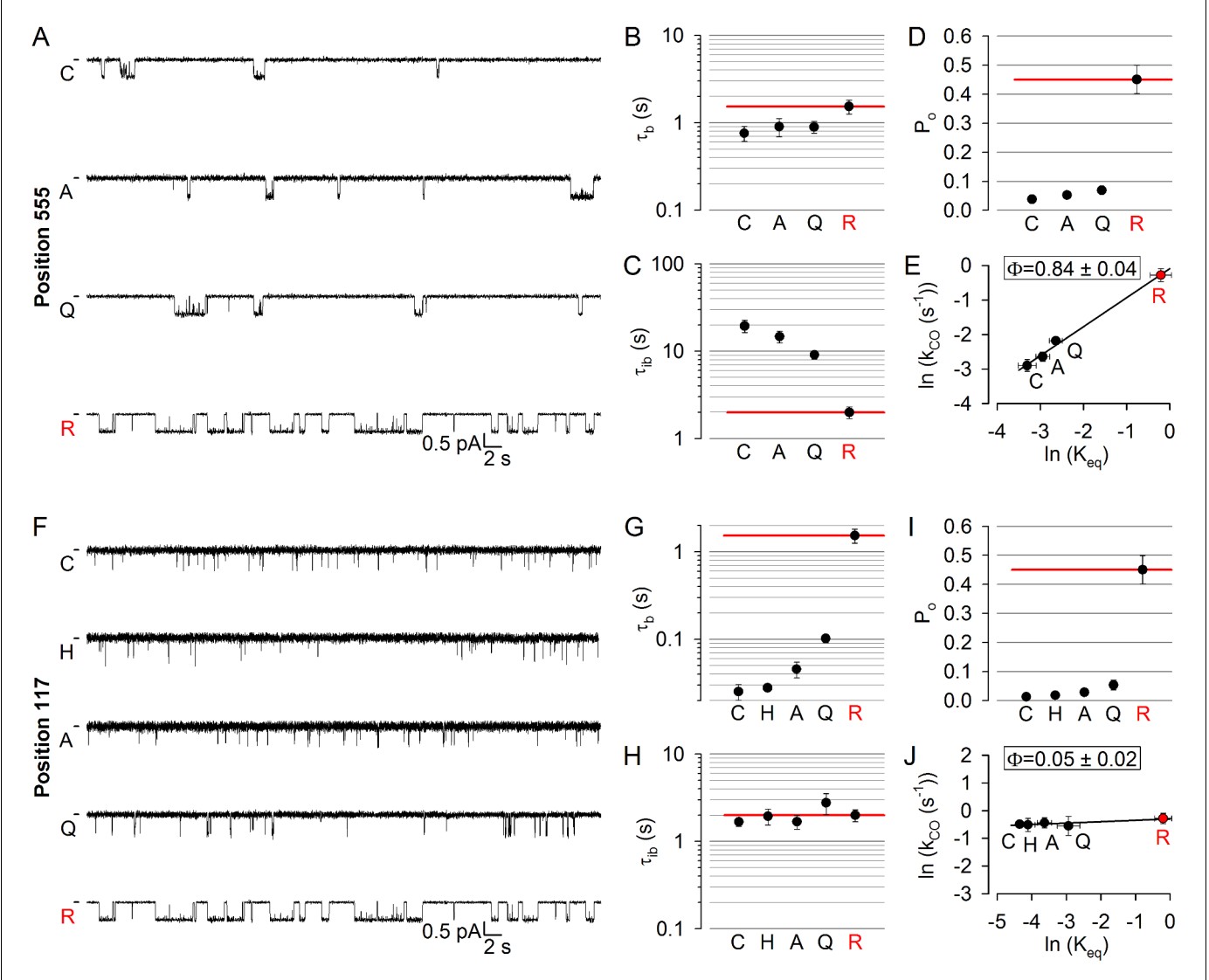

**Figure 2.** Longitudinal Φ value gradient extends from catalytic site 2 to the extracellular surface. (A, F) Inward unitary currents of the cut-ΔR(D1370N) CFTR background construct, and of channels bearing mutations at either position 555 (A) or position 117 (F), in the same background. *Letters* to the left of the traces indicate the amino acid present in the target position; the native residue is marked by *red*. Currents were recorded in symmetrical 140 mM Cl⁻, at −80 mV in (A) but at −100 mV in (F); *dashes* on the left mark zero-current level. (B–D) and (G–I), Mean burst (B, G, $\tau_b$) and interburst (C, H, $\tau_{ib}$) durations and open probabilities (D, I, $P_o$) of the constructs in *A* and *F*, respectively. *Red horizontal lines* highlight the respective control values of the background construct which is identified by the *red letter* representing the native target residue. All data are shown as mean ± SEM (n = 6 for data in B–D, n = 5–7 for data in G–I). (E, J) Brønsted plots for position 555 (E) and 117 (J). *Red symbol* and *letter* identifies the background construct. *Solid lines* are linear regression fits with slope Φ indicated.

DOI: https://doi.org/10.7554/eLife.29013.003

The following figure supplements are available for figure 2:

**Figure supplement 1.** All tested constructs are saturated by 10 mM ATP.
DOI: https://doi.org/10.7554/eLife.29013.004
**Figure supplement 2.** Stimulation of open probability by 2'-deoxy-ATP (dATP) or N⁶-(2-phenylethyl)-dATP (P-dATP) facilitates counting channels for low-$P_o$ mutants.
DOI: https://doi.org/10.7554/eLife.29013.005
**Figure supplement 3.** Burst analysis of single-channel recordings for position 555 mutants.
DOI: https://doi.org/10.7554/eLife.29013.006
**Figure supplement 4.** Burst analysis of single-channel recordings for position 117 mutants.
DOI: https://doi.org/10.7554/eLife.29013.007

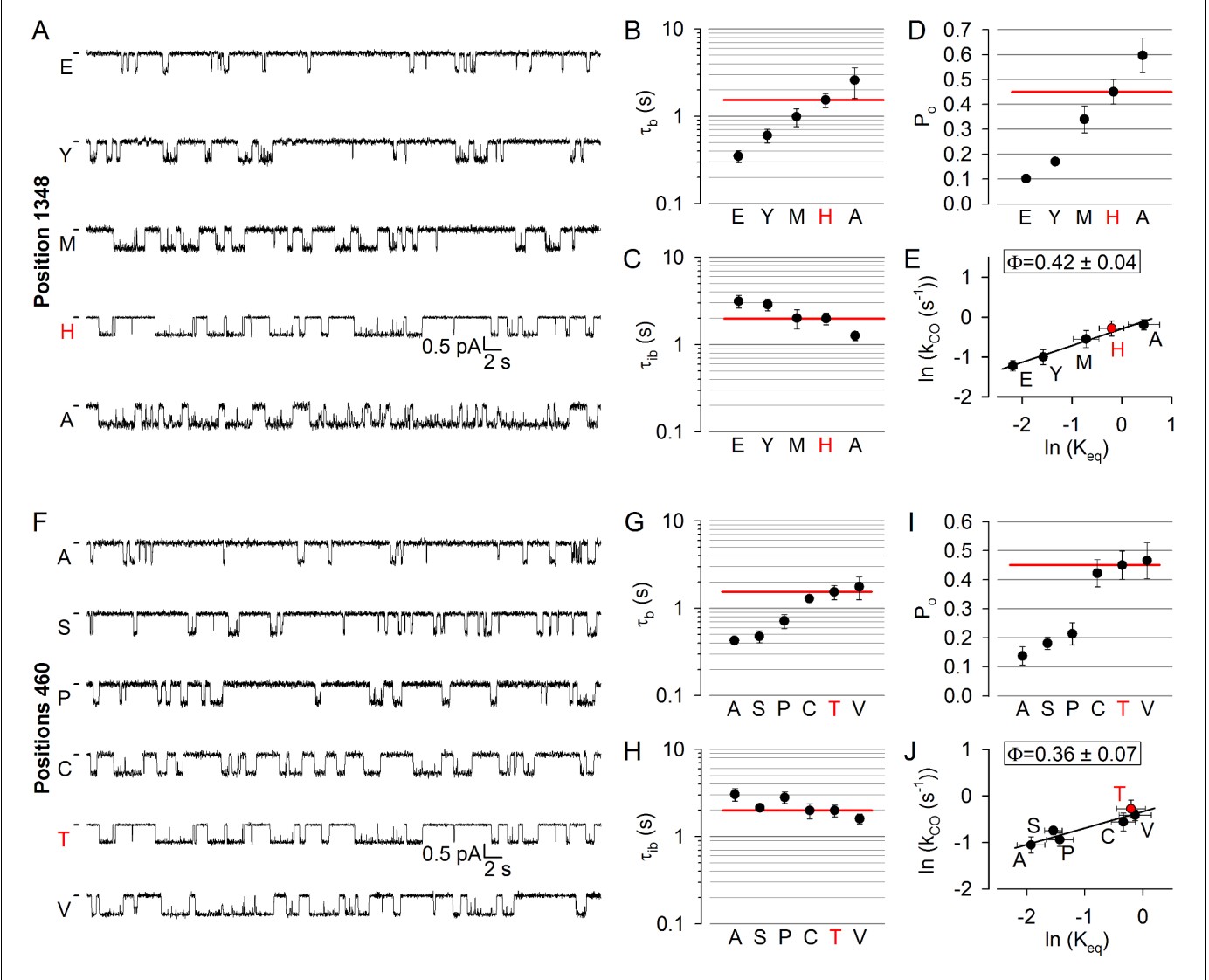

**Figure 3.** Delayed movement in non-catalytic site 1 during pore opening. (A, F) Inward unitary currents of the cut-ΔR(D1370N) CFTR background construct, and of channels bearing mutations at either position 1348 (A) or position 460 (F), in the same background. *Letters* to the left of the traces indicate the amino acid present in the target position; the native residue is marked by *red*. Currents were recorded at −80 mV, in symmetrical 140 mM Cl⁻; *dashes* on the left mark zero-current level. (B–D) and (G–I), Mean burst (B, G, $\tau_b$) and interburst (C, H, $\tau_{ib}$) durations and open probabilities (D, I, $P_o$) of the constructs in A and F, respectively. *Red horizontal lines* highlight the respective control values of the background construct which is identified by the *red letter* representing the native target residue. All data are shown as mean ± SEM (n = 5–10 for data in *B-D*, n = 5–10 for data in *G–I*). (E, J) Brønsted plots for position 1348 (E) and 460 (J). *Red symbol* and *letter* identifies the background construct. *Solid lines* are linear regression fits with slope Φ indicated.

DOI: https://doi.org/10.7554/eLife.29013.008

The following figure supplements are available for figure 3:

**Figure supplement 1.** Burst analysis of single-channel recordings for position 1348 mutants.

DOI: https://doi.org/10.7554/eLife.29013.009

**Figure supplement 2.** Burst analysis of single-channel recordings for position 460 mutants.

DOI: https://doi.org/10.7554/eLife.29013.010

$P_o$ (*Figure 3F,I*). As with position 1348, these changes in $P_o$ again reflected coupled opposing changes in mean burst and interburst durations (*Figure 3G,H*; cf., histograms in *Figure 3—figure supplement 2*). Correspondingly, the Brønsted plot for position 460 (*Figure 3J*) revealed a low-intermediate value of Φ = 0.36 ± 0.07, corroborating the conclusion from position 1348 that

movement within non-catalytic site 1 during pore opening is markedly delayed compared to that within catalytic site 2. Importantly, although perturbations of the NBD1 Walker A motif might impair ATP binding at non-catalytic site 1 (*Vergani et al., 2003*), 10 mM ATP remained saturating for all of our site-1 mutants (*Figure 2—figure supplement 1A*; *Figure 2—figure supplement 1B*, *orange bars*).

### No time-asymmetry in opening-related motions is detectable at the level of the four coupling helices

The large $\Phi$ of positions in catalytic site 2 but low-to-intermediate $\Phi$ of positions in non-catalytic site 1 implies that timing of movements in the two ATP sites during pore opening is highly asymmetric: whereas catalytic site 2 has already adopted its open-like conformation in the T state, non-catalytic site 1 is still on the move. If the movements completed in non-catalytic site 1 between the T state and open state are large-scale rearrangements, then a similarly pronounced temporal asymmetry of motions might be detectable one structural level further along the axis, at the level of the four coupling helices (*Figure 1C*, *violet helices*), since CH1 and CH2 are located closer to non-catalytic site 1, whereas CH3 and CH4 closer to catalytic site 2 (*Figure 1D*, *violet helices*).

To test for such possible asynchrony, we targeted a position in each of the four coupling helices: 172 in CH1, 275 in CH2 (studied earlier; [*Sorum et al., 2015*]), 961 in CH3, and 1068 in CH4 (*Figure 1A,C–D*, *purple*). At positions 172, 961, and 1068 (the native residues are leucine, methionine, and phenylalanine, respectively) substitutions caused both increases and decreases in open probability (*Figure 4D,I,N*), and the underlying changes in gating pattern were similar (*Figure 4A,F, K*): a clear trend for an association of lengthened bursts with shortened interbursts and of shortened bursts with lengthened interbursts could be observed (compare Figs. B-C, G-H, L-M; cf., histograms in *Figure 4—figure supplements 1–3*). Thus, perturbations at positions 172, 961, and 1068 affect opening and closing rates in opposite directions but to comparable extents, yielding Brønsted plots with intermediate slopes in each case (*Figure 4E,J,O*), indicating that in the T state the coupling helices have already left their closed-like, but have not yet reached their final open-like, positions. On the other hand, the obtained values of $\Phi = 0.51 \pm 0.07$ for position 172, $\Phi = 0.60 \pm 0.12$ for position 961, and $\Phi = 0.54 \pm 0.07$ for position 1068 are quite similar to each other and to $\Phi = 0.50 \pm 0.13$ for position 275 (*Sorum et al., 2015*).

### Disease hotspot position 508 is on the move in the opening transition state

Position 508 (*Figure 1A,C–D*, *deep purple*), the locus of the most common (>70%) CF mutation, is found in a surface cleft of NBD1 which accommodates CH4 to form a joint-like interface in which the F508 side chain interacts with aromatic CH4 residues, including F1068 (*Zhang and Chen, 2016*). Deletion of phenylalanine 508 is detrimental for folding and stability of the CFTR protein (*Du et al., 2005*; *Okiyoneda et al., 2010*), but also severely impairs gating (*Miki et al., 2010*; *Kopeikin et al., 2014*). To address timing of motions in this disease hotspot-position, we replaced the native phenylalanine with cysteine, leucine, and serine. All substitutions markedly reduced open probability (*Figure 5A,D*), supporting exquisite sensitivity of this region to structural perturbations. The reduction in the closed-open equilibrium constant was caused in each case by a simultaneous shortening of mean burst and lengthening of mean interburst durations, which were comparably affected (*Figure 5B–C*; cf., histograms in *Figure 5—figure supplement 1*). The resulting $\Phi = 0.55 \pm 0.02$ (*Figure 5E*) is indistinguishable from that of position 1068, and indicates that position 508 is also just on the move in the T state for opening. Again, importantly, the reduced opening rate of the position 508 mutants was not due to impaired ATP binding, as 10 mM ATP remained saturating (*Figure 2—figure supplement 1B*, *deep purple bars*). Channels could be efficiently counted at the end of each experiment by exposure to dATP or P-dATP (*Figure 2—figure supplement 2*).

## Discussion

The closed-to-open transition of the CFTR channel is a major structural rearrangement which involves dimerization of the NBDs and flipping of the TMDs from an inward- to an outward-facing conformation ([*Vergani et al., 2005*], cf., homology models for phosphorylated closed and open CFTR channels in *Figure 6A*, *left*-to-*right*). Whereas X-ray or cryo-EM structures may capture stable

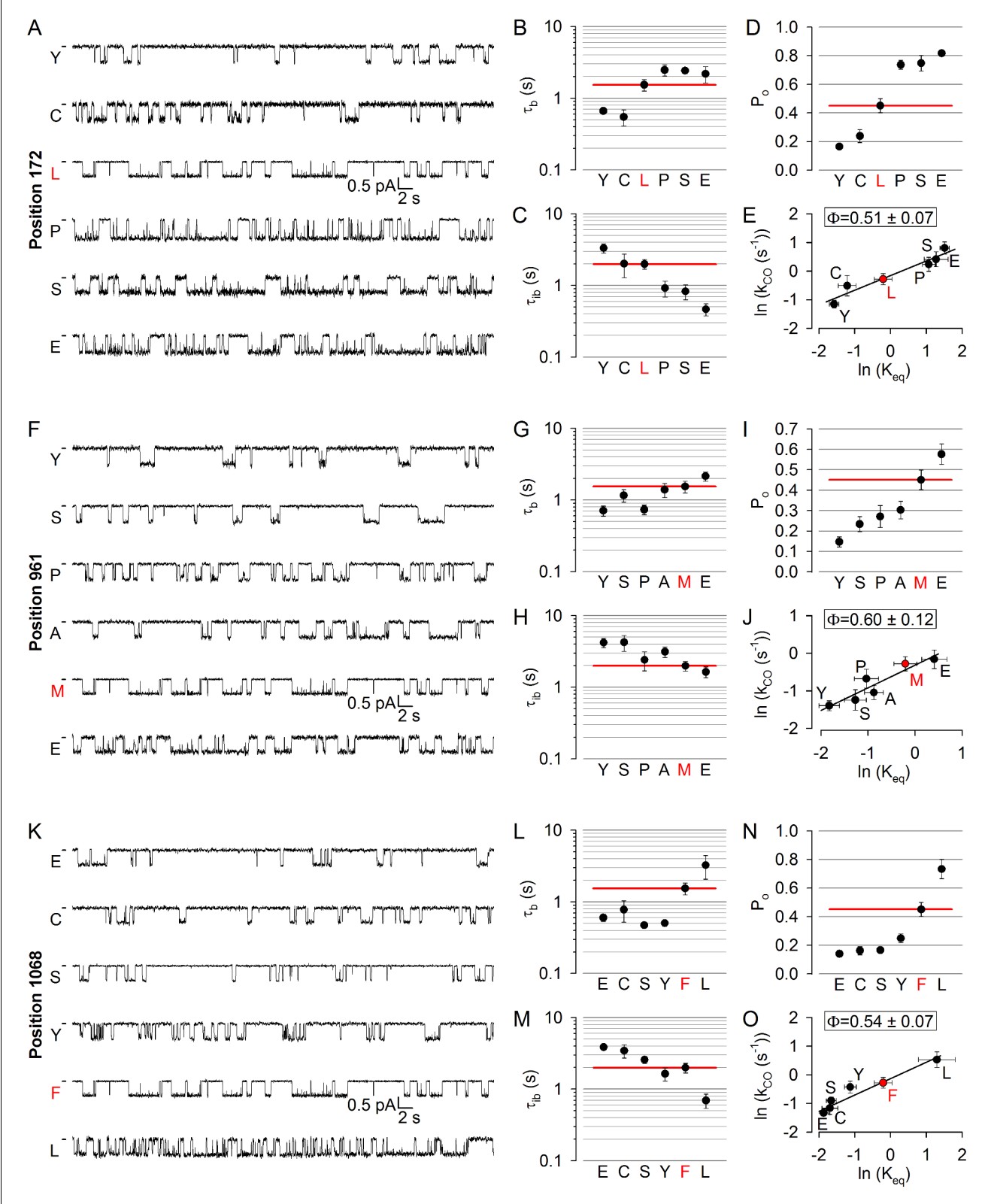

**Figure 4.** No asymmetry in the timing of motions can be detected at the level of the coupling helices. (A, F, K) Inward unitary currents of the cut-ΔR (D1370N) CFTR background construct, and of channels bearing mutations at position 172 (A), 961 (F), or 1068 (K), in the same background. *Letters* to the left of the traces indicate the amino acid present in the target position; the native residue is marked by *red*. Currents were recorded at −80 mV, in symmetrical 140 mM Cl⁻; *dashes* on the left mark zero-current level. (B–D, G-I) and (L–N), Mean burst (B, G, L, $\tau_b$) and interburst (C, H, M, $\tau_{ib}$) durations

*Figure 4 continued*

and open probabilities (D, I, N, P$_o$) of the constructs in A, F, and K, respectively. *Red horizontal lines* highlight the respective control values of the background construct which is identified by the *red letter* representing the native target residue. All data are shown as mean ± SEM (n = 3–8 for data in (B–D), n = 4–15 for data in *G-I*, n = 4–7 for data in L–N). (E, J, O) Brønsted plots for position 172 (E), 961 (J), and 1068 (O). *Red symbol* and *letter* identifies the background construct. *Solid lines* are linear regression fits with slope Φ indicated.

DOI: https://doi.org/10.7554/eLife.29013.011

The following figure supplements are available for figure 4:

**Figure supplement 1.** Burst analysis of single-channel recordings for position 172 mutants.

DOI: https://doi.org/10.7554/eLife.29013.012

**Figure supplement 2.** Burst analysis of single-channel recordings for position 961 mutants.

DOI: https://doi.org/10.7554/eLife.29013.013

**Figure supplement 3.** Burst analysis of single-channel recordings for position 1068 mutants.

DOI: https://doi.org/10.7554/eLife.29013.014

closed (*Zhang and Chen, 2016*; *Liu et al., 2017*) and open ground states, the structure of the transition state traversed during opening, and the relative timing of motions in various protein regions may presently be inferred only from Φ value analysis (*Auerbach, 2007*). Previous work revealed a Φ value gradient along the longitudinal protein axis, suggesting that the opening conformational change spreads from catalytic site 2 towards the pore (*Sorum et al., 2015*). By expanding the axial distance spanned by our target positions here we show a gradient of Φ values from ~1 for both sides of catalytic ATP site 2, all the way to ~0 for the extracellular protein surface (*Figure 6A*). These findings corroborate the earlier proposal that in the T state catalytic site 2 is already dimerized but the pore is still in its closed-like (inward-facing) conformation.

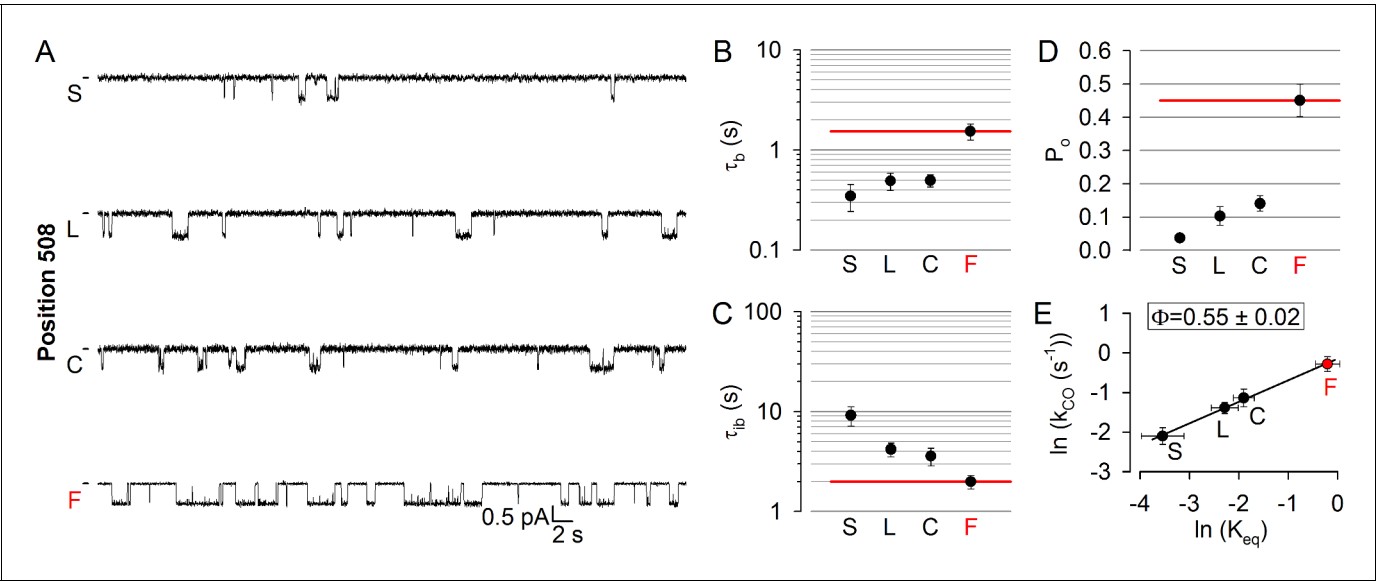

**Figure 5.** Disease hotspot position 508 is on the move in the transition state for opening. (A), Inward unitary currents of the cut-ΔR(D1370N) CFTR background construct, and of channels bearing mutations at position 508, in the same background. *Letters* to the left of the traces indicate the amino acid present in the target position; the native residue is marked by *red*. Currents were recorded at −80 mV, in symmetrical 140 mM Cl$^-$; *dashes* on the left mark zero-current level. (B–D) Mean burst (B, τ$_b$) and interburst (C, τ$_{ib}$) durations and open probabilities (D, P$_o$) of the constructs in (A). *Red horizontal lines* highlight the respective control values of the background construct which is identified by the *red letter* representing the native target residue. All data are shown as mean ± SEM (n = 5–7). E, Brønsted plot for position 508. *Red symbol* and *letter* identifies the background construct. *Solid line* is a linear regression fit with slope Φ indicated.

DOI: https://doi.org/10.7554/eLife.29013.015

The following figure supplement is available for figure 5:

**Figure supplement 1.** Burst analysis of single-channel recordings for position 508 mutants.

DOI: https://doi.org/10.7554/eLife.29013.016

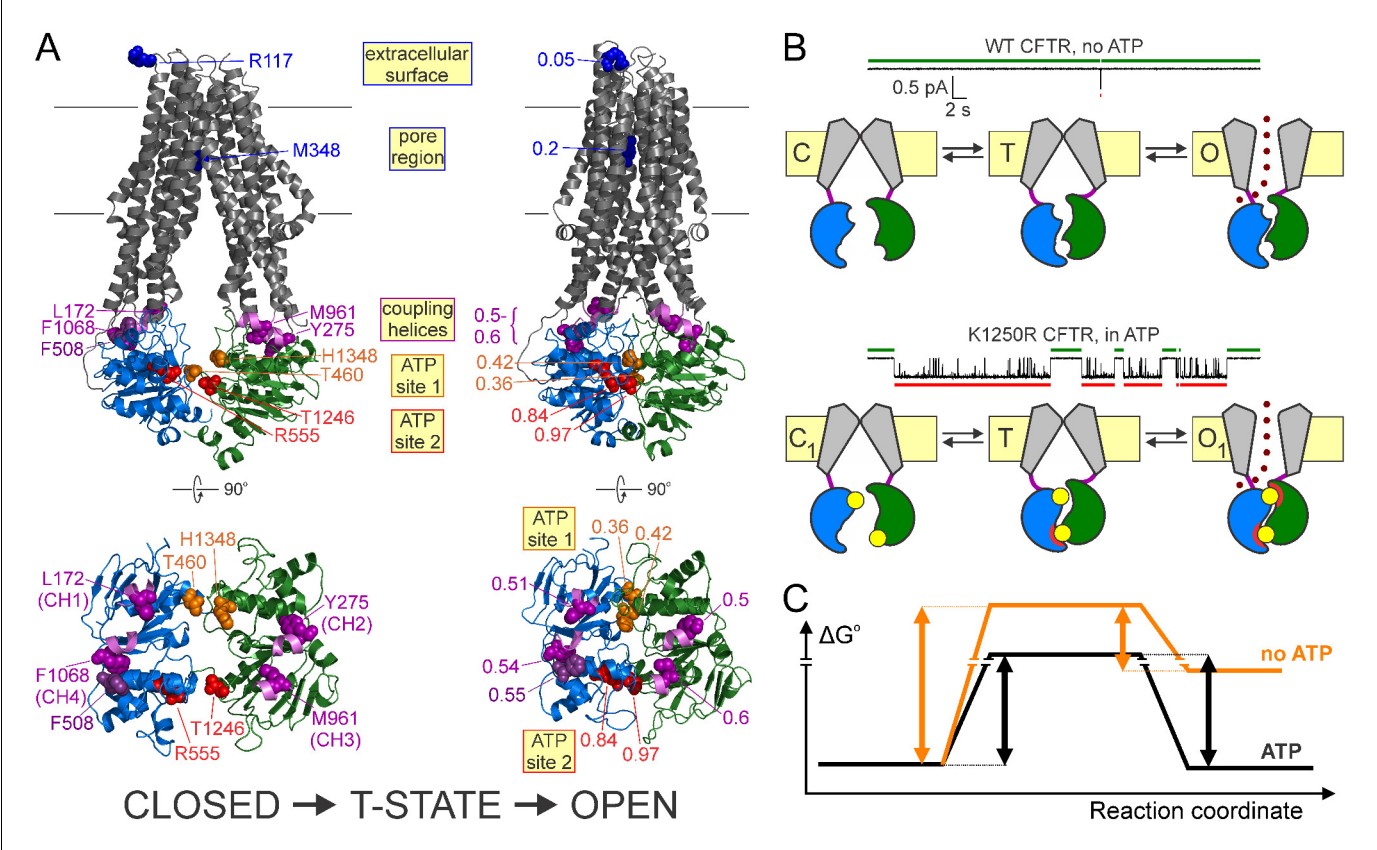

**Figure 6.** Distinct roles of the two ATP sites in promoting CFTR channel gating. (**A**) Homology models (*Corradi et al., 2015*) of closed (*left*) and open (*right*) conformations of phosphorylated CFTR gating in ATP, based on the structures of inward-facing Tm287-288 (*left*) and outward-occluded McjD (*right*); target positions are highlighted in spacefill. Labels identify the native residues (*left*) and the Φ values of each position (*right*). For both conformations full structures (*top*) and NBDs with coupling helices only (*bottom*) are shown. (**B**) Current traces showing a single pore opening event in the absence of ATP in a patch containing hundreds of WT CFTR channels (*top trace*), and gating of a single K1250R CFTR channel in 5 mM ATP (*bottom trace*). *Green* and *red bars* identify closed interburst and open burst events, respectively. Cartoons below the traces illustrate the mechanism of gating of WT CFTR in the absence (*top*) and of K1250R CFTR in the presence (*bottom*) of ATP. Color coding as in *Figure 1B*, the *upper* ATP binding site represents non-catalytic site 1, the *lower* site represents catalytic site 2. *Red semi-circles* around ATP (*yellow circles*) represent tight bonding of the nucleotide with the opposing NBD surface. (**C**) Standard Gibbs energy profiles of a CFTR channel in the absence (*orange profile*) and presence (*black profile*) of bound ATP during progress from the closed state (*left*) through the transition state (*center*) to the open state (*right*). *Vertical arrows* illustrate the heights of the energetic barriers for the forward (opening) and backward (closing) steps under both conditions.

DOI: https://doi.org/10.7554/eLife.29013.017

The following figure supplement is available for figure 6:

**Figure supplement 1.** Location of aspartate 1370, mutated in our background construct.
DOI: https://doi.org/10.7554/eLife.29013.018

Phenylalanine 508, the locus of the most common CF mutation, is located at the NBD-TMD interface, that is, in a region expected to experience strain in the T state. Here we show that position 508 is indeed just on the move in the T state, together with spatially adjacent CH4 (*Zhang and Chen, 2016*; *Liu et al., 2017*), as reported by their intermediate, essentially indistinguishable Φ values of ~0.5 (*Figures 4O* and *5E*). Because all tested perturbations of the F508 side chain reduce opening and accelerate non-hydrolytic closing rate (*Figure 5B–C*), the simplest interpretation is that the native phenylalanine side chain is involved in stabilizing interactions that become stronger as the closed channel reaches the T state, and even stronger when it relaxes to the open state. Loss (or weakening) of these interactions by 508-position perturbations destabilizes the T state relative to the closed state (reducing opening rate, *Figure 5C*; [*Miki et al., 2010*]), and further destabilizes the open state relative to the T state (enhancing non-hydrolytic closing rate, *Figure 5B*; [*Jih et al., 2011*]). The alternative explanation that the native phenylalanine side chain forms destabilizing

interaction(s) in the closed state seems unlikely, as mutations here dramatically reduce protein stability, suggesting an overall stabilizing role for residue F508 in its native environment. Of note, because in wild-type (WT) CFTR the hydrolytic pathway for channel closure is much faster than the non-hydrolytic pathway, the strong acceleration of non-hydrolytic closing rate by deletion or substitutions of residue F508 is not apparent when studied in a WT (hydrolytic) background (*Miki et al., 2010*; *Kopeikin et al., 2014*), a fact that has led to previous misassignment of a Φ value close to 1 for position 508 (*Aleksandrov et al., 2009*).

Mutation R117H is associated with a mild form of CF and causes, beside a moderate reduction in single channel chloride conductance, a strong channel gating defect (*Sheppard et al., 1993*; *Yu et al., 2016*), suggesting that this position is involved in interactions that change in a gating-state dependent manner. Indeed, all substitutions at position 117 dramatically reduced open probability (*Figure 2I*) by accelerating closure (*Figure 2G*) while leaving opening unaffected (*Figure 2H*). Because in closed-state structures of CFTR the native R117 side chain is not seen to form strong interactions with other residues (*Zhang and Chen, 2016*; *Liu et al., 2017*), the implication is that it forms a stabilizing interaction (or interactions) in the open state, that is (are) lost when the arginine at position 117 is replaced. Furthermore, the stabilizing interaction is not yet formed in the T state: its appearance is among the last movements associated with pore opening (Φ ~ 0). Thus, R117 substitutions selectively destabilize the open state, but not the transition state, relative to the closed state, as indicated by the unaltered opening rate.

In striking contrast to the large Φ values of positions on both sides of the dimer interface in catalytic site 2, positions flanking non-catalytic site 1 display low-to-intermediate Φ values (*Figure 6A*, *orange* vs. *red residues* (*left*) and *numbers* (*right*)). Thus, unlike catalytic site 2 which has already adopted its open-like conformation (*Figure 6A*, *right*, *red residues*) in the T state, non-catalytic site 1 is still on the move between its closed- and open-like conformations (*Figure 6A*, *left* and *right*, *orange residues*). Interestingly, such pronounced asymmetry in the timing of motions at one end of the dimer interface compared to the other cannot be detected at the level of the four coupling helices, which are all characterized by Φ values of 0.5–0.6 (*Figure 6A*, *violet residues* (*left*) and *numbers* (*right*)). Thus, the movements that take place in non-catalytic site 1 between the T state and the open state likely remain confined to the site-1 interface.

ATP bound at the two composite sites acts as molecular glue that stabilizes the open-pore conformation by bonding the NBD interfaces together (*Moody et al., 2002*; *Smith et al., 2002*). Our data suggest a division of labor in the bonding activities of the two sites. Phosphorylated WT CFTR channels open infrequently even in the absence of ATP (*Figure 6B*, *top current trace*), and such 'spontaneous' openings reflect occasional dimerization of empty, unliganded NBDs (*Mihályi et al., 2016*) (*Figure 6B*, *top cartoon*). Comparison of such spontaneous gating of WT channels with the equilibrium gating of a hydrolysis-deficient mutant, such as the NBD2 Walker A lysine mutant K1250R, in saturating ATP (*Figure 6B*, *bottom current trace*) reveals two robust effects of bound ATP: a shortening of closed interburst durations (*Figure 6B*, compare *green bars* above current traces) and a lengthening of open burst durations (*Figure 6B*, compare *red bars* below current traces), both by >100 fold (*Mihályi et al., 2016*). Thus, bound ATP reduces the energetic barrier for opening, but increases the barrier for non-hydrolytic closure (*Figure 6C*, *black* vs. *orange* standard Gibbs energy profiles).

Which of the two bound ATP molecules affects opening rate? Our Φ value analysis indicates that in the T state catalytic site 2 has already finished moving, that is, the site-2 glue is already bonded (*Figure 6B*, *bottom cartoon*, *lower site*; *red semi-circles* in states T and $O_1$): that bonding readily explains the reduction in the energetic barrier for opening, that is, stimulation of opening rate, by ATP. Whether stably bound ATP in non-catalytic site 1 might also contribute to speeding opening, by maintaining some contact across the site-1 interface even in the closed state and hence preventing complete NBD disengagement, is still unsettled (*Tsai et al., 2010b*; *Szollosi et al., 2011*), but, cf. *Chaves and Gadsby, 2015*).

Which of the two bound ATP molecules affects non-hydrolytic closing rate? Because it is bonded both in the T and $O_1$ states, the site-2 glue cannot affect the height of the barrier for non-hydrolytic closure. On the other hand, the low Φ value of non-catalytic site 1 indicates localized movements between the T state and the open state, which we propose reflects tight bonding of the site-1 glue (i.e., tightening of the site-1 interface; *Figure 6B*, *bottom cartoon*, *upper site*; *red semi-circle* only in state $O_1$). By selectively stabilizing the open state relative to the T state, the site-1 glue increases the

height of the barrier for non-hydrolytic closure, explaining the observed prolongation of non-hydrolytic bursts by ATP.

In the context of the functional cycle of WT CFTR (*Figure 1B*) ATP bound at catalytic site 2 promotes efficient opening (speeds step $C_1 \rightarrow O_1$) whereas ATP bound at non-catalytic site 1, by slowing rate $O_1 \rightarrow C_1$, ensures progress through ATP hydrolysis: such unidirectional cycling is essential for ABC proteins to mediate uphill transport. In asymmetric ABC proteins flipping to the outward-facing, NBD-dimerized conformation is followed by a step in which the consensus site becomes committed to ATP hydrolysis (*Timachi et al., 2017*): that step might involve bonding of the glue in the non-catalytic site. This interpretation is consistent with the functional cross-talk between the two sites observed in CFTR or related MRP1 (*Ramjeesingh et al., 1999*; *Gao et al., 2000*; *Hou et al., 2002*): perturbations in the non-catalytic site of CFTR reduce the ATPase turnover rate of the catalytic site (*Ramjeesingh et al., 1999*) by lowering the fraction of open bursts that are terminated by ATP hydrolysis as opposed to non-hydrolytic dissociation of the NBD dimer (*Csanády et al., 2010*).

The hydrolysis-disrupting D1370N mutation, an essential component of our background construct, precludes targeting position 1370 for $\Phi$ value analysis. However, the non-hydrolytic closing rate of D1370N CFTR channels is substantially faster than that of other non-hydrolytic catalytic site-2 mutants, suggesting that this mutation affects not only opening, but also non-hydrolytic closing rate (*Gunderson and Kopito, 1995*; *Vergani et al., 2003*): the implication is that the $\Phi$ value of position 1370 might be lower than those of the catalytic site-2 positions studied here. Although functionally involved in $Mg^{2+}$ coordination in catalytic site 2 (cf., *Zaitseva et al., 2005*), aspartate 1370 is physically located in between the two ATP sites, in the center of the NBD interface, at the boundary between core and $\alpha$-helical subdomains of NBD2 (*Figure 6—figure supplement 1*, *salmon*). Moreover, it is in contact with the conserved Q-loop glutamine Q1291 (*Figure 6—figure supplement 1*, *yellow*) (*Liu et al., 2017*) which acts as a $\gamma$-phosphate sensor to induce an ~15° rotation of the $\alpha$-helical subdomain towards the core subdomain upon ATP binding (*Karpowich et al., 2001*). Thus, residue D1370 might also be involved in subdomain closure, essential for stable NBD dimerization (*Smith et al., 2002*). Whether in CFTR that subdomain closure already happens in closed channels upon ATP binding (state $C_1$, *Figure 1B*), or is completed only in the tight-dimer open-channel state as in the maltose transporter (*Orelle et al., 2010*), might be clarified by establishing a different non-hydrolytic background construct which allows targeting positions at the intra-NBD subdomain interface for $\Phi$ value analysis.

In conclusion, the $\Phi$ value map obtained here supports distinct roles for the two composite ATP sites in CFTR channel gating and reveal the molecular mechanism of the gating defects caused by two common CF mutations. Whereas this work reports relative *timing* of movements in the two ATP sites during pore opening, clarifying the *extent* of these movements during each step in the channel opening/closing cycle will require comparisons among high-resolution structures of CFTR trapped in phosphorylated closed- and open-channel states. The recent structures of unphosphorylated closed (*Liu et al., 2017*) and phosphorylated occluded (*Zhang et al., 2017*) CFTR have just started to set the stage for such comparisons.

## Materials and methods

### Molecular biology

Mutations at positions 117, 172, 460, 508, and 555 were introduced into pGEMHE-CFTR(1-633) (*Csanády et al., 2000*), and those at positions 961, 1068, and 1348 into pGEMHE-CFTR(837-1480 (D1370N)) (*Sorum et al., 2015*) using Stratagene QuikChange (Agilent Technologies, Santa Clara, CA, USA). cDNA was transcribed in vitro using T7 polymerase. Particular substitutions were chosen based on the following general considerations: (i) mutations that are too conservative might not cause a phenotype (and hence result in a dot on the REFER plot which lies very close to that of the WT), whereas (ii) mutations that are too drastic might disrupt functional expression. Because predicting the impact of a particular substitution is often difficult, for each position typically 5–6 point substitutions were chosen in a quasi-arbitrary manner, ranging from conservative mutations to substitutions that markedly alter side chain size, electrostatic charge, chemical nature, or backbone conformation (proline). All mutants that gave rise to functional channels were characterized.

## Isolation and injection of Xenopus laevis oocytes

Ovarian lobes were removed from anaesthetized *Xenopus laevis* [RRID:NXR_0.0080] following a IACUC-approved protocol. Oocytes were defolliculated by treatment with Type II collagenase (Sigma-Aldrich, Hungary) and stored at 18°C in a frog Ringer's solution supplemented with 1.8 mM $CaCl_2$ plus 50 μg/ml gentamycin sulfate (Sigma). Recordings were done 1–3 days after co-injection with 0.1–10 ng cRNA for both CFTR segments.

## Single-channel patch-clamp recording

CFTR unitary currents were recorded in excised inside-out patches at 25°C as described earlier (*Sorum et al., 2015*). Pipette solution contained (in mM) 136 NMDG-Cl, 2 $MgCl_2$, 5 HEPES, pH = 7.4 with NMDG, bath solution contained 134 NMDG-Cl, 2 $MgCl_2$, 5 HEPES, 0.5 EGTA, pH = 7.1 with NMDG. The bath solution was continuously flowing, and could be exchanged with a time constant of ~100 ms. MgATP (3 or 10 mM) and 2'-deoxy-ATP sodium salt (dATP, 5 mM) (Sigma) were added from 400 mM and 100 mM aqueous stock solutions, respectively (pH = 7.1 with NMDG); dATP was supplemented with equimolar $MgCl_2$. 25–50 μM 2'-deoxy-$N^6$-(2-phenylethyl)-ATP sodium salt (P-dATP; Biolog LSI, Germany) was added from a 10 mM aqueous stock solution. Unitary CFTR currents in 10 mM MgATP were recorded at −80 mV (−100 mV for position 117 mutants) (Axopatch 200B, Molecular Devices, Sunnyvale, CA, USA) digitized at 1 kHz (Digidata 1322A, Pclamp9 [RRID:SCR_011323], Molecular Devices), and lowpass-filtered at 50 Hz using a Gaussian filter. Single-channel patches were identified as very long (typically 15 min - 1 hr) recordings without superimposed channel openings. For the low-$P_o$ mutants targeting positions 117, 508, and 555 strong stimulation by dATP or P-dATP at the end of each experiment was used to facilitate correct estimation of the number of active channels in the patch (*Figure 2—figure supplement 2*). Macroscopic or quasi-macroscopic current ratios between 3 and 10 mM ATP were used to verify saturation by 10 mM ATP for each of the tested mutants (*Figure 2—figure supplement 1*). For display purposes figure panels show channel currents filtered at 20 Hz (50 Hz for R117 mutants).

## Data analysis

Bursts and interbursts were reconstructed as described earlier (*Sorum et al., 2015*). In brief, currents from long segments of recording without superimposed channel openings were filtered at 50 Hz, idealized by half-amplitude threshold crossing, and brief closures suppressed using the method of (*Magleby and Pallotta, 1983*). Opening ($k_{CO}$) and closing ($k_{OC}$) rates were defined as the inverses of the mean interburst ($\tau_{ib}$) and burst ($\tau_b$) durations, respectively, and $K_{eq}$ as $k_{CO}/k_{OC}$. Plots of $\ln(k_{CO})$ as a function of $\ln(K_{eq})$ (Brønsted plots) were fitted by linear regression (SigmaPlot 8 [RRID:SCR_003210]).

## Statistics

All data are given as mean ± SEM of measurements from ≥3 (typically 5–7, as indicated in each figure legend) long segments of single-channel recordings, from 3 to 15 patches for each mutant. In the face of alternating periods of lower and higher activity typical to CFTR (*Bompadre et al., 2005*), several hours of total recording for each construct were obtained to ensure unbiased sampling of average gating behaviour, and all data were included in the analysis.

## Acknowledgements

We thank David Gadsby and Jue Chen for critical review and discussions. Supported by MTA Lendület grants LP2012-39/2012 and LP2017-14/2017, Research Grant CSANAD17G0 from the Cystic Fibrosis Foundation, and an International Early Career Scientist grant from the Howard Hughes Medical Institute to LC.

## Additional information

### Competing interests
László Csanády: Reviewing Editor, eLife. The other authors declare that no competing interests exist.

### Funding

| Funder | Grant reference number | Author |
|---|---|---|
| Howard Hughes Medical Institute | International Early Career Scientist Award | László Csanády |
| Cystic Fibrosis Foundation | Research Grant CSANAD17G0 | László Csanády |
| Magyar Tudományos Akadémia | Lendület grant LP2017-14/2017 | László Csanády |
| Magyar Tudományos Akadémia | Lendület grant LP2012-39/2012 | László Csanády |

The funders had no role in study design, data collection and interpretation, or the decision to submit the work for publication.

### Author contributions
Ben Sorum, Data curation, Formal analysis, Investigation; Beáta Töröcsik, Resources, Methodology; László Csanády, Conceptualization, Software, Supervision, Funding acquisition, Validation, Methodology, Writing—original draft, Project administration

### Author ORCIDs
Ben Sorum http://orcid.org/0000-0001-6742-1094
László Csanády http://orcid.org/0000-0002-6547-5889

### Ethics
Animal experimentation: This study was performed in strict accordance with the recommendations in the Guide for the Care and Use of Laboratory Animals of the National Institutes of Health. All of the animals were handled according to approved institutional animal care and use committee (IACUC) protocols of Semmelweis University (last approved 06-30-2016, expiration 06-30-2021).

### Decision letter and Author response
Decision letter https://doi.org/10.7554/eLife.29013.020
Author response https://doi.org/10.7554/eLife.29013.021

## Additional files

### Supplementary files
• Transparent reporting form
DOI: https://doi.org/10.7554/eLife.29013.019

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
