## [Decision Letter]

Thank you for submitting your article "Asymmetry of movements in CFTR's two ATP sites during pore opening serves their distinct functions" for consideration by *eLife*. Your article has been favorably evaluated by Richard Aldrich (Senior Editor) and three reviewers, one of whom, Kenton J Swartz (Reviewer #1), is a member of our Board of Reviewing Editors. The following individual involved in review of your submission has agreed to reveal their identity: Roderick MacKinnon (Reviewer #3).

The reviewers have discussed the reviews with one another and the Reviewing Editor has drafted this decision to help you prepare a revised submission.

Summary:

This interesting manuscript from the Csanady lab presents new results continuing work started in a previous paper analyzing the open/closed transition state of the CFTR chloride channel. This channel, related to the ABC transporter family, undergoes a highly complex gating process which requires channel phosphorylation in the R domain as well as ATP binding and hydrolysis at the NBDs to drive irreversible gating. Though much progress has occurred in understanding these processes, key questions remain open. Here the authors dissect the final gating step by using transition state analysis of a mutant that is claimed to introduce a simple, two-state open closed reaction for the opening of the channel. Such a two-state system is required for the analysis to be valid. The authors analyze the effects of mutations at a series of positions on the open/closed equilibrium that complement the previous set, focusing now on areas around the ATP1 binding site and the outside of the pore. They find that the timing of the opening wave reveals propagation from ATP2 to ATP1, which is roughly simultaneous with movement of the interfacial coupling helices. A position at the outside end of the pore moves only very late in the reaction. Notably, the timing differences between the ATP binding sites seem not to persist through the interfacial helices, which all seem to move nearly simultaneously. The work reported here is careful and thorough, and the single channel recordings are really nice. The presentation is also excellent, very clearly discussing a difficult topic. This kind of work is highly complementary to the spate of emerging structures and provides information that will be difficult to obtain using other methods. We do have several concerns that the authors should address before we are fully convinced by their arguments. The authors should be aware that the reviewers have somewhat divergent opinions about the most interesting new conclusions in this manuscript.

Essential revisions:

1) The single-step assumption. Critical to the argument presented here is that the equilibrium is a single step reaction, and that the transitions we observe are between two discrete states. However, our impression is that the gating reflects the binding of ATP (at least to site 2), dimerization of the NBDs, and the channel conformational change. From the collective Phi's measured in this and the previous paper, and their apparently smooth gradation from slopes of near zero to near 1, we could be convinced that this is a single reaction, but we think the burden is on the authors to substantiate this a bit more. How single exponential are the results? Are *all* the mutants equally well fit by a single tau each? The reported tau values are said to be from single exponential fits, but I think it would be worthwhile to establish this formally. Perhaps the authors could show the dwell time histograms (or at least examples) and fits in a supplemental figure.

2) We are unclear of the data in the second part of Figure 3 and its associated discussion (paragraph beginning "To target the opposing…"). Please explain with greater care what you have done. Specifically, we would rather not see pooled data sets (from 460 and 464). Since the most important result in this paper is the discovery of different Phi values for catalytic and non-catalytic sites, we would be more confident in seeing what the Phi values are for multiple positions within each site (the theory that Phi values are related to a wave of conformational change across the protein, we should expect to see multiple similar Phi values in a given region.) Further, we are puzzled by the minimal discussion of the complexity of the data in Figure 3. Because the different Phi values between the two sites is the main result of this paper, we need to be more convinced of the data.

3) We think the authors should be more circumspect in their interpretation about how Phi analysis relates to timing of movements. The analysis speaks to where in the transition mutants at specific sites cause a perturbation. It may be reasonable to infer from this that this result is because those regions undergo dynamic rearrangements at different times in the transition, but it would seem more appropriate to separate the specific conclusion from the inference.

4) We were somewhat confused by the cartoon in Figure 6 and its implications for the gating model. Our understanding is that ATP1 binds stably and rarely exchanges, where ATP2 binding and hydrolysis drives the reaction. In this light, isn't it surprising that the movements at ATP1 occur *after* those of ATP1? I guess the model in our head is that the NBDs are partially 'glued' by ATP1 and ready to close upon ATP2 binding. Is this incorrect? If not, what is happening to ATP1 that makes the movements around it happen later?

---

## [Author Response]

Essential revisions:1) The single-step assumption. Critical to the argument presented here is that the equilibrium is a single step reaction, and that the transitions we observe are between two discrete states. However, our impression is that the gating reflects the binding of ATP (at least to site 2), dimerization of the NBDs, and the channel conformational change.

Because opening indeed requires binding of ATP to site 2 followed by the channel isomerization step, it is critical that all experiments be performed in saturating ATP to make the isomerization step rate limiting. We have verified that this condition is met for all of our mutants (Figure 2—figure supplement 1).

From the collective Phi's measured in this and the previous paper, and their apparently smooth gradation from slopes of near zero to near 1, we could be convinced that this is a single reaction, but we think the burden is on the authors to substantiate this a bit more. How single exponential are the results? Are all the mutants equally well fit by a single tau each? The reported tau values are said to be from single exponential fits, but I think it would be worthwhile to establish this formally. Perhaps the authors could show the dwell time histograms (or at least examples) and fits in a supplemental figure.

For our background construct we have demonstrated the validity of the single-step assumption in our earlier study (Sorum et al., 2015). In a series of new supplementary figures (Figure 2—figure supplement 3–Figure 2—figure supplement 4; Figure 3—figure supplement 1–Figure 3—figure supplement 2; Figure 4—figure supplement 1–Figure 4—figure supplement 3; Figure 5—figure supplement 1) we now illustrate the entire procedure of our burst analysis for two mutants per each studied position. For those positions at which different substitutions caused bi-directional changes in P_o_ (1348, 172, 961, 1068) the chosen examples represent the mutants at the two extreme ends of the REFER plot.

Panels B (G) and C (H) show distributions of simple closed and open times, and the choice of the burst delimiter (t_crit_) according to the algorithm of Magleby and Pallotta (1983). Panels D (I) and E (J) display the distributions of the durations of *reconstructed* bursts and interbursts, resulting from suppression of brief "flickery" closures, and reasonable fits of the distributions of *reconstructed* bursts by single-exponential probability density functions. We note, however, that the distributions of *reconstructed* interburst durations will never be truly single-exponential: first, the left tail of the *true* interburst distribution is truncated, and second, the right tail of the distribution of flickery closures will survive – this is particularly apparent for the high-P_o_ mutants for which the two closed-time components overlap substantially (e.g., H1348A, L172E, M961E, F1068L). However, the choice of t_crit_ assures that the mean duration of *reconstructed* interburst events matches that of the *true* interburst events (and therefore the estimation of mean burst durations is also unbiased). We would like to point out that the values of τ_b_ and τ_ib_ were not derived from single-exponential fits (the *reconstructed* interburst distributions are not single-exponential), but reflect simple arithmetic averages of burst and interburst dwell-time durations (see Materials and methods). All this is explained in the legend of the first histogram figure (Figure 2).

2) We are unclear of the data in the second part of Figure 3 and its associated discussion (paragraph beginning "To target the opposing…"). Please explain with greater care what you have done.

Our original aim was to characterize Φ for position 464, which has been extensively studied by several groups in the past. However, unfortunately most of the mutants we generated failed to form functional channels, with the exception of K464C and K464R for which we did manage to obtain a few recordings. On the assumption that Φ values for nearby positions will be similar, we therefore sought to solidify the K464 data by combining them with data obtained from two mutants of nearby position 460 (the position analogous to 1246 in site 2). We acknowledge that this approach depends on the above assumption.

Specifically, we would rather not see pooled data sets (from 460 and 464). Since the most important result in this paper is the discovery of different Phi values for catalytic and non-catalytic sites, we would be more confident in seeing what the Phi values are for multiple positions within each site (the theory that Phi values are related to a wave of conformational change across the protein, we should expect to see multiple similar Phi values in a given region.)

Done. Because we were unable to obtain a reliable Φ value for position 464, we have omitted the 464 data altogether. Instead, we have generated and functionally characterized three additional 460-position mutants, and now provide a clean REFER plot for position 460. The obtained Φ value of 0.36 ± 0.07 for position 460 is only slightly different from the Φ obtained earlier for the pooled 460-464 data (0.26 ± 0.06), and is quite similar to that of position 1348 located at the opposing surface of site 1 (0.42 ± 0.04). Thus, our conclusions are now based on the Φ values of two positions within each of the two ATP sites.

Further, we are puzzled by the minimal discussion of the complexity of the data in Figure 3. Because the different Phi values between the two sites is the main result of this paper, we need to be more convinced of the data.

The complexity of the notation (each construct denoted by a pair of letters) was due to the pooled nature of the original data set – in the present version this problem has been eliminated.

3) We think the authors should be more circumspect in their interpretation about how Phi analysis relates to timing of movements. The analysis speaks to where in the transition mutants at specific sites cause a perturbation. It may be reasonable to infer from this that this result is because those regions undergo dynamic rearrangements at different times in the transition, but it would seem more appropriate to separate the specific conclusion from the inference.

We now clarify this point in the Introduction:

"For the opening step of a simple idealized two-state channel with a single intervening T state, Φ is a measure of how far the conformation of the position has progressed along the reaction coordinate in the T state (0≤Φ≤1). […] The terms "early" and "late" will be used here to reflect sequentiality of movements based on this interpretation."

4) We were somewhat confused by the cartoon in Figure 6 and its implications for the gating model. Our understanding is that ATP1 binds stably and rarely exchanges, where ATP2 binding and hydrolysis drives the reaction. In this light, isn't it surprising that the movements at ATP1 occur after those of ATP1?

We do not think that there is anything surprising about this finding. It is clear that opening is initiated by ATP binding to site 2, because channels immediately cease to reopen once ATP is removed from the bath solution, even though site 1 likely keeps ATP bound for some tens of seconds (Tsai et al., 2010). The mere fact that opening is *initiated* by site 2 necessarily means that site 2 must undergo some movement first, before site 1 also reacts: causality must imply sequentiality, otherwise site 1 would have to be able to "look into the future", to anticipate ATP binding-induced rearrangements in site 2.

I guess the model in our head is that the NBDs are partially 'glued' by ATP1 and ready to close upon ATP2 binding. Is this incorrect?

Indeed, data from the Hwang laboratory (Tsai et al., 2010) and from our group (Szollosi et al., 2011) have provided evidence for some interaction across site 1 which is maintained even in the closed state ("partially glued NBDs"). If true, then this effect might also contribute to speeding opening (relative to unliganded gating) by preventing the NBDs from distancing themselves from each other as far as seen in unliganded cryo-EM structures (Zhang et al., 2016; Liu et al., 2017). However, this issue is still unsettled, as work from the Gadsby laboratory (Chaves and Gadsby, 2015) suggested complete separation of both ATP sites during each channel closing event.

We have now added a sentence to the Discussion to briefly expose this controversial topic: "Whether stably bound ATP in non-catalytic site 1 might also contribute to speeding opening, by maintaining some contact across the site-1 interface even in the closed state and hence preventing complete NBD disengagement, is still unsettled (Tsai et al., 2010b; Szollosi et al., 2011, but, cf. Chaves and Gadsby, 2015)."

Furthermore, we would like to point out that it is not "ATP2 *binding*" that initiates opening: in the closed channel ATP binding/unbinding to site 2 is a rapid equilibrium process, whereas opening rate is extremely slow (~1 s^-1^) even in saturating ATP. Thus, the movement at site 2 which initiates opening must occur *after* ATP2 binding, and has been shown to reflect closure of the site-2 NBD interface (Vergani et al., 2003; Vergani et al., 2005).

If not, what is happening to ATP1 that makes the movements around it happen later?

Regardless of whether site 1 separation is complete or incomplete in the closed state, the causality argument discussed above does not leave much room for any alternative explanation. The exact nature of these movements is obviously beyond the scope of our study.